# Identification of Determinants of Biofeedback Treatment’s Efficacy in Treating Migraine and Oxidative Stress by ARIANNA (ARtificial Intelligent Assistant for Neural Network Analysis)

**DOI:** 10.3390/healthcare10050941

**Published:** 2022-05-19

**Authors:** Irene Ciancarelli, Giovanni Morone, Maria Giuliana Tozzi Ciancarelli, Stefano Paolucci, Paolo Tonin, Antonio Cerasa, Marco Iosa

**Affiliations:** 1Department of Life, Health and Environmental Sciences, University of L’Aquila, 67100 L’Aquila, Italy; irene.ciancarelli@univaq.it (I.C.); mariagiuliana.tozzi@univaq.it (M.G.T.C.); 2Santa Lucia Foundation IRCSS, 00179 Roma, Italy; s.paolucci@hsantalucia.it (S.P.); marco.iosa@uniroma1.it (M.I.); 3S. Anna Rehabilitation Institute, RAN-Research on Advanced Neurorehabilitation, 88900 Crotone, Italy; p.tonin@isakr.it (P.T.); antonio.cerasa@irib.cnr.it (A.C.); 4Institute for Biomedical Research and Innovation (IRIB), National Research Council of Italy, 98164 Messina, Italy; 5Pharmacotechnology Documentation and Transfer Unit, Preclinical and Translational Pharmacology, Department of Pharmacy, Health Science and Nutrition, University of Calabria, 87036 Rende, Italy; 6Department of Psychology, Sapienza University of Rome, 00185 Rome, Italy

**Keywords:** artificial intelligence, artificial neural network, oxidative stress, migraine, headache, nitric oxide, superoxide dismutase, biofeedback

## Abstract

Migraines are a public health problem that impose severe socioeconomic burdens and causes related disabilities. Among the non-pharmacological therapeutic approaches, behavioral treatments such as biofeedback have proven effective for both adults and children. Oxidative stress is undoubtedly involved in the pathophysiology of migraines. Evidence shows a complex relationship between nitric oxide (NO) and superoxide anions, and their modification could lead to an effective treatment. Conventional analyses may fail in highlighting the complex, nonlinear relationship among factors and outcomes. The aim of the present study was to verify if an artificial neural network (ANN) named ARIANNA could verify if the serum levels of the decomposition products of NO—nitrite and nitrate (NOx)—the superoxide dismutase (SOD) serum levels, and the Migraine Disability Assessment Scores (MIDAS) could constitute prognostic variables predicting biofeedback’s efficacy in migraine treatment. Twenty women affected by chronic migraine were enrolled and underwent an EMG-biofeedback treatment. The results show an accuracy for the ANN of 75% in predicting the post-treatment MIDAS score, highlighting a statistically significant correlation (R = −0.675, *p* = 0.011) between NOx (nitrite and nitrate) and MIDAS only when the peroxide levels in the serum were within a specific range. In conclusion, the ANN was proven to be an innovative methodology for interpreting the complex biological phenomena and biofeedback treatment in migraines.

## 1. Introduction

Migraines are one of the most common diseases worldwide and a major cause of disability, with a substantial social burden [1]. The effects on the daily lives of migraineurs and their families are heavy, and are amplified because the prevalence of migraines is the highest in midlife, when work and family productivity, responsibilities, and demands are pressing [2]. The burden of migraines is significantly high, with greater headache-related disability and decrements in health-related quality of life in addition to necessitating a notable utilization of healthcare resources and imposing higher direct and indirect costs [3]. Biofeedback is an established non-pharmacological technique commonly used in the treatment of migraines, proving effective for decreasing the severity of, frequency of, and disability resulting from chronic headaches and migraines [4]. The effectiveness of biofeedback in limiting migraines is linked to the muscular relaxation induced by reducing affective stress and also by the modulation of oxidative stress, a phenomenon recognized as characterizing migraine patients [5]. Behavioral treatments have been proven to produce significant results regarding classical primary endpoints such as headache frequency and secondary endpoints such as mood disorders, disability, and quality of life, acting through the modulation of muscle contracture, relaxation, changes in biomarkers, and the inhibition of cortical excitability, thereby reducing cortical spreading depression [6]. However, even though the use of biofeedback is supported by strong scientific evidence, the National Institute for Health and Care Excellence (NICE) of the United Kingdom does not recommended non-pharmacological treatments because most of the studies lacked control groups and large sample sizes [7]. The majority of the clinical studies show the efficacy of behavioral approaches for headache treatment. In migraine prophylaxis, biofeedback shows the same effectiveness as pharmacological treatments, with additional effects when this approach is used in combination with pharmacological treatments. Therefore, behavioral approaches are useful in patients who cannot tolerate or are non-responsive to preventive or acute drugs, or in children [8,9].

Nitric oxide (NO) is a non-adrenergic, non-cholinergic neurotransmitter processing noxious impulses and sensitizing perivascular sensory nerves [10]. NO rapidly reacts with superoxide anions because of their unpaired electrons in the outer orbitals, resulting in a rapid radical/radical reaction that reduces its effective half-life and biological function [11]. In addition, the interaction between NO and superoxide anions causes the production of peroxynitrite, a strong oxidant that influences the mechanism of hyperalgesia in pain and mediates the dilation of cerebral arterioles through an oxidant mechanism occurring during migraine attacks. Therefore, the interaction between NO and superoxide anions may be involved in inducing and maintaining migraine-related changes in cerebral blood flow [12]. Superoxide dismutase (SOD) is a family of metalloenzymes that catalyzes the dismutation of superoxide anions into molecular oxygen and hydrogen peroxide, and it is an essential component of the cellular antioxidant defense mechanism. SOD reduces the superoxide concentration and prevents NO’s decomposition into nitrite and nitrate (NOx) by scavenging superoxide anions. In migraine sufferers, the activity of the radical-scavenging enzyme SOD is lower than that in healthy controls, suggesting a decreased effectiveness of antioxidant defenses and an enhanced vulnerability to oxidative stress [13].

Furthermore, the management of migraines with cognitive behavioral therapy often carries a very low risk of side effects, and the treatments are often well accepted by migraine sufferers who do not want to follow drug therapy, avoiding the possible side effects [14]. Biofeedback is a technique that has proven to be effective in the modulation of oxidative stress and SOD bioavailability [5], as well as in inducing, as a main action, the control and modification of some of the body’s functions, such as the muscle contractures typical of migraineurs. During a biofeedback session, patients are connected to electrical sensors that monitor selected body functions such as muscle tension, and based on this feedback, the relaxation of certain muscles is promoted, which may reduce migraine frequency and pain severity [15].

A priority need in clinical practice is identifying prognostic factors for the efficacy of therapeutic approaches. To this end, in consideration of the complexity of the clinical mechanisms underlying migraines, it was considered useful and appropriate to evaluate the assessed data from our previous study regarding the use of biofeedback in the modulation of oxidative stress in migraineurs [4], with an innovative and powerful methodology such as artificial neural networks (ANNs). Neural networks reflect the behavior of the human brain, allowing computer programs to recognize patterns and solve problems in the medical field. Neural networks rely on training data to learn and improve their accuracy over time. However, once these learning algorithms are fine-tuned for accuracy, they are powerful tools in computer science and artificial intelligence, allowing us to classify and cluster data at a high velocity with a good level of accuracy.

ANNs are sets of nonlinear data computational models consisting of input and output layers as well as one or more hidden layers. Inspired by the human nervous system, the artificial neural network can recognize patterns, manage data, and, most significantly, learn. This learning ability, absent from other computer models simulating human intelligence, constantly improves its functional accuracy as it continues to function. Noteworthy is its effectiveness in classifying and interpreting the various forms of medical data that helps clinical decision making in both diagnosis and treatments [15,16,17,18,19].

Given this complex scenario, the aim of the present study was to investigate whether the ANN analysis of the values of NO bioavailability, SOD activity, and MIDAS scores assessed before biofeedback treatment might allow the consideration of these variables as predictors of biofeedback’s efficacy in migraineurs.

## 2. Materials and Methods

The authors declare that the present study was a secondary analysis of data already acquired in a previous study by Ciancarelli et al. that evaluated the relationship between biofeedback treatment’s efficacy and the modulation of oxidative stress in patients with chronic migraine [5].

### 2.1. Participants, Biomarkers, and Migraine Assessment

Twenty women (mean age: 25.7 ± 3.7 years) with chronic migraines diagnosed according to the International Classification of Headache Disorders, 2nd Edition criteria [20] were enrolled. The inclusion and exclusion criteria utilized for the patients’ enrollment are specified in the previous paper of Ciancarelli et al. [5]. No pharmacological preventive treatment was allowed. Paracetamol (1000 mg) was selected as an acute medication for treating migraine attacks, as it does not appear to modify oxidative stress in humans. Blood samples were collected to analyze the nitrite and nitrate (NOx), SOD, and peroxide levels; the first and second blood samples were taken on two different days during a headache-free period, corresponding to the day of the first biofeedback session and that of the last one, respectively.

Extensive and complete information regarding the blood samples’ collection and processing is presented in the previous manuscript [5]. The migraine frequency, severity, and disability were evaluated according to the Migraine Disability Assessment Score (MIDAS) [21], administered before and soon after the scheduled biofeedback sessions. The MIDAS is a brief questionnaire designed to quantify headache-related disability and is comprised of five questions that are evaluated to obtain a score that represents the disability severity: grade I, little or no disability (scores 0 to 5); grade II, mild disability (scores 6 to 10); grade III, moderate disability (scores 11 to 20); grade IV, severe disability (scores 21 or higher).

### 2.2. Biofeedback Treatment

Disposable cup-type electrodes were applied to the frontal muscle: the active electrodes were directly centered over each eye, while the reference electrode was centered directly over the bridge of the nose. The operator adjusted the threshold of the auditory feedback signal provided to the patient from session to session depending on the level of muscular tension. The operator’s task was to successively shape lower EMG levels; the subject’s task was to go below the threshold and turn off the auditory signal completely. In this way, subjects gradually learned to reduce their muscle tension. Migraine sufferers underwent three consecutive sessions of biofeedback per week for 12 sessions, with at least a one day interval between sessions. The type of biofeedback used and further information regarding the training adopted can be found in the paper of Ciancarelli et al. [5].

### 2.3. Artificial Neural Network

We used an artificial intelligence network (ANN) based on an artificial neural network already developed in other studies called ARIANNA (ARtificial Intelligent Assistant for Neural Network Analysis) (Figure 1) [17,18,19]. ARIANNA is a multilayer perceptron, formed by the input layer, two hidden layers, and a final output layer (the output of which was the predicted outcome). The architecture of the ARIANNA was that of a feed-forward neural network (FFNN), with data moving in only one direction, from the input nodes through the two hidden layers to the output node. The activation function for all the units in the hidden layers and for the output layer was a hyperbolic tangent. The chosen computational procedure was based on online training (online training uses information from one record at a time, updating the weights until one of the stopping rules is met [17,18,19]). Differently from previous studies [17,18,19], the number of hidden units was not fixed a priori but was automatically determined by the artificial neural network. The input layers referred to the following variables assessed pre-treatment: age, SOD, NOx, peroxides, and MIDAS. The output layer provided the estimation of the outcome measure that was the MIDAS assessed post-treatment. The ANN was implemented using the specific toolbox Neural Networks of IBM SPSS Statistics version 23 (Armonk, NY: IBM Corp).

### 2.4. Statistical Analysis

The data are reported in terms of the means ± standard deviations. The normality of the data distribution was assessed by using the Shapiro–Wilk test. A paired t-test was used to compare data pre- and post-treatment. The Pearson coefficient (R) was used to assess correlations. The importance and normalized importance of the input variables of the artificial neural network were computed.

## 3. Results

No dropouts occurred during the study period, nor were outliers present in the data. No patients took acute medication for treating migraine attacks.

Table 1 reports the values of the assessed parameters pre- and post-treatment, with their paired comparison performed according to the normality check. Pre-treatment, the MIDAS was found to be significantly correlated with peroxide (R = 0.451, *p* = 0.046), and partially so with SOD (R = −0.380, *p* = 0.098), but not with NO (R = −0.008, *p* = 0.972). The MIDAS assessed post-treatment was correlated neither with the pre-treatment values nor with the post-treatment values (the only significant correlation was between the MIDAS pre- and post-treatment: R = 0.863, *p* < 0.001).

Figure 1 shows a schematic representation of ARIANNA. The numbers of elements self-determined for the two hidden layers were 20 and 15, respectively. Thirteen values of the MIDAS post-treatment were perfectly predicted (accuracy = 65%), and another two values showed an error <5 (cumulative accuracy = 75%). As shown in Figure 2, in four out of the five remaining cases there was an overestimation of the MIDAS post-treatment.

Table 2 reports the importance (weight) for each one of the input variables in determining the output. We found that NO was the most important one. There was not a general correlation between the pre-treatment NO and post-treatment MIDAS (R = −0.079, *p* = 0.741), but if the peroxides were in the range 116–205, the correlation between the NO and MIDAS was statistically significant (R = −0.675, *p* = 0.011).

## 4. Discussion

Chronic migraineurs have higher oxidative stress and a lower antioxidant capacity [5,22,23], and the expression of nitrate, nitrite, and nitric oxide reductase genes is significantly higher in migraineurs than in non-migraineurs [24]. In line with these assumptions, our previous results showed, in chronic migraine before biofeedback sessions, decreased SOD and NOx serum levels and increased peroxide serum levels with respect to the levels in healthy control subjects [5]. In our previous study, the lower NO bioavailability in migraineurs was explained as a consequence of decreased SOD activity, which probably caused a quicker and more consistent reaction of NO with free-radical species such as peroxides, decreasing the NO level. These data were confirmed by the lack of significant differences in the NOx serum levels, as well as SOD and peroxide levels, between the migraineurs after biofeedback and healthy control subjects [5]. In this study, the ANN analysis validated the efficacy of biofeedback in limiting oxidative processes by improving SOD activity and thus scavenging superoxide anions (Table 1), underlying the role of biofeedback training not only as an efficacious behavioral/relaxation therapy, but also as a strategic treatment to reduce the vulnerability of migraineurs to oxidative stress [5,25]. Moreover, the muscular relaxation induced by biofeedback is promoted by an enhancement of NO bioavailability through the activation of NO pathways [5,26]. Furthermore, the relaxation-based treatment performed with biofeedback is confirmed to be extremely useful as a therapeutic approach, decreasing the headache-related disability and improving the independence in the activities of daily living of migraineurs, as determined by the ANN analysis of the MIDAS score, which significantly decreased after the biofeedback sessions, suggesting the potential effectiveness of biofeedback in migraine treatment, as well as in migraineurs abusing analgesic drugs and who have greater compliance with non-pharmacological treatments. [5,25,26] The results of our study are in line with the conclusions of the most recent manuscripts, confirming the efficacy of behavioral approaches in headache treatment. [2,4,6,14] Particularly, our results also show that biofeedback, inducing muscular relaxation and modulating biomarkers, represents an efficacious non-pharmacological approach for migraine prophylaxis, as also described in other manuscripts. [9,14,15]

The main limitation of this study was the small sample size of 20 participants; for each, five variables were assessed at baseline. Further studies should investigate more samples. However, despite this small dataset, the ANN achieved good accuracy in predicting the outcome.

With the purpose of interpreting the complex relationship emerging from biological data, as in our study, in order to identify prognostic factors for migraine recovery and disability alleviation, the ANN turns out to be an innovative methodology able to highlight important relationships that simple correlations may fail to identify as statistically significant. The complexity of these relationships was, in fact, confirmed by the absence of significant correlations of the pre-treatment variables with the post-treatment MIDAS score. The complexity of these relationships also results from the high number of hidden elements (20 and 15) self-determined by the neural network and needed to predict the outcome. The most important factor useful for predicting the MIDAS score post-treatment was the NOx serum levels, followed by the peroxide serum levels (both assessed pre-treatment). As stated above, there was not a simple linear correlation between these latter two variables and the MIDAS post-treatment; for this reason, there was a need for a more complex algorithm to highlight their influence on this outcome. The accuracy of the ANN was about 75%, with an overestimation of the MIDAS score in most of the remaining cases (25%), as also shown by the frequency distributions reported in Figure 2. The ANN results suggest that a higher level of NO pre-treatment is related to a lower MIDAS score post-treatment, but only if peroxides are in a specific range (116–205 U/mL) excluding extreme values pre-treatment (lower than 116 U/mL or higher than 205 U/mL). For this reason, data analysis with the ARIANNA methodology, despite the high complexity of the neural network (with a total of 35 hidden elements), constitutes a significant opportunity in clinical practice for identifying prognostic factors for the efficacy of therapeutic approaches.

## 5. Conclusions

The analysis conducted by using the artificial neural network ARIANNA on the relationship between biomarkers and biofeedback treatment in migraineurs revealed a complex relationship in which the increase in NOx, when serum level of peroxides lies within a specific range, is the most important factor for predicting biofeedback’s efficacy in reducing migraines. In conclusion, the perspective of this study is to reiterate the efficacy of biofeedback in the prophylactic treatment of migraines and, above all, to underline that the analysis of biological data with the ANN may represent an appropriate methodology for identifying the predictive factors for therapeutic effectiveness.

## Figures and Tables

**Figure 1 healthcare-10-00941-f001:**
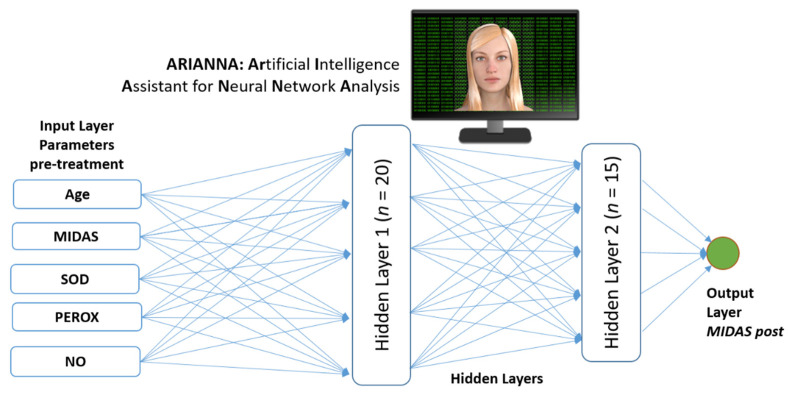
Schematic representation of the artificial neural network used in this study with five variables assessed pre-treatment as input and one variable assessed post-treatment (MIDAS) as output (*n* represents the number of units determined for each hidden layer).

**Figure 2 healthcare-10-00941-f002:**
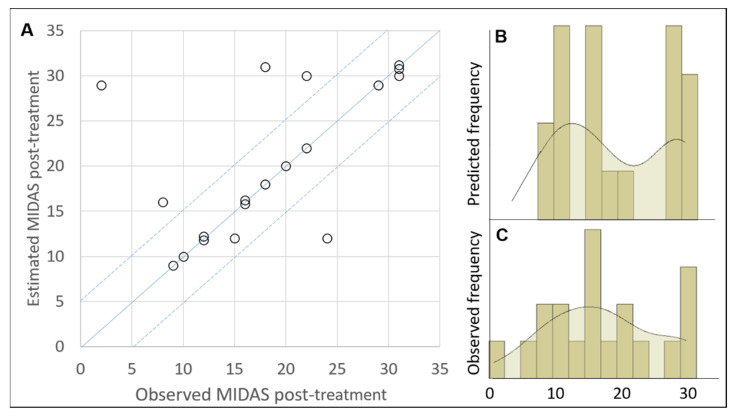
The results of the artificial neural network. Panel (**A**) Observed vs. estimated MIDAS post-treatment; the blue solid line represents the perfect prediction; the dotted line represents a tolerance of 5. Panel (**B**) Frequency distribution of predicted MIDAS. Panel (**C**) Frequency distribution of observed MIDAS.

**Table 1 healthcare-10-00941-t001:** Means ± standard deviations of the parameters assessed pre- and post-treatment for the group of participants, with the *p*-values of the paired comparison and normality check.

Assessment of Variables	Pre-Treatment	Post-Treatment	Paired Comparison T-Test, *p*-Value	Normality Shapiro–Wilk Test, *p*-Value
SOD (μM)	6.5 ± 1.0	8.0 ± 0.7	<0.001	0.372
NOx (μM)	23.7 ± 4.2	31.4 ± 3.0	<0.001	0.612
Peroxides (U/mL)	145.8 ± 40.3	82.5 ± 21.3	<0.001	0.199
MIDAS	37.0 ± 13.2	18.8 ± 8.6	<0.001	0.102

**Table 2 healthcare-10-00941-t002:** Results of ARIANNA with the weights associated with each pre-treatment variable for determining the MIDAS post-treatment as the outcome.

Pre-Treatment Variable	Importance in the ANN	Normalized Importance
Age (years)	0.184	83.4%
SOD (μM)	0.189	85.6%
NOx (μM)	0.221	100%
Peroxides (U/mL)	0.216	97.9%
MIDAS	0.191	86.5%

## Data Availability

Not applicable.

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
