# Peer review of "Identification of Determinants of Biofeedback Treatment’s Efficacy in Treating Migraine and Oxidative Stress by ARIANNA (ARtificial Intelligent Assistant for Neural Network Analysis)"

_healthcare, 2022, doi:10.3390/healthcare10050941_

Round 1
Reviewer 1 Report
This study investigates whether the ANN analysis of the values of NO bioavailabilty, SOD activity and MIDAS scores assessed before biofeedback treatment may allow to consider these variables as predictors of the biofeedback efficacy in magraineurs patients. The results showed that there was not a general correlation between pre-treatment NO and post-treatment MIDAS, but if the peroxides are in the range, the correlation between NO and MIDAS was statistically significant. They conclude that there is a complex relationship in which the increased of NOx within a specific range of peroxides serum, is the most important factor to predict biofeedback efficacy in reducing migraine.
My major concern is below;
All of the patients enrolled in the study are chronic migraineurs and they may already use preventive treatment and acute medication for the attacks. However, there is no information about the drugs which patients used, these drugs may also effect the serum level of oxidant and antioxidant levels. It would be better to include this information in the methods section and discuss this in regard with the results and serum oxidant/antioxidant levels in the manuscript.
Author Response
REV 1
This study investigates whether the ANN analysis of the values of NO bioavailabilty, SOD activity and MIDAS scores assessed before biofeedback treatment may allow to consider these variables as predictors of the biofeedback efficacy in magraineurs patients. The results showed that there was not a general correlation between pre-treatment NO and post-treatment MIDAS, but if the peroxides are in the range, the correlation between NO and MIDAS was statistically significant. They conclude that there is a complex relationship in which the increased of NOx within a specific range of peroxides serum, is the most important factor to predict biofeedback efficacy in reducing migraine.
My major concern is below;
All of the patients enrolled in the study are chronic migraineurs and they may already use preventive treatment and acute medication for the attacks. However, there is no information about the drugs which patients used, these drugs may also effect the serum level of oxidant and antioxidant levels. It would be better to include this information in the methods section and discuss this in regard with the results and serum oxidant/antioxidant levels in the manuscript.
We thanks the Reviewer for the observation that allows us to specify that no pharmacological preventive treatments was prescribed during the study period (only 1 month) just not modify the serum level of oxidant and antioxidant levels as well as the possible effectiveness of biofeedback treatment. About the acute medication for migraine attacks treatment, paracetamol (1000 mg) was allowed and has been selected as it seems to not induce modification in oxidative stress in humans (Trettin A, Böhmer A, Suchy MT, Probst I, Staerk U, Stichtenoth DO, Frölich JC, Tsikas D. Effects of Paracetamol on NOS, COX, and CYP Activity and on Oxidative Stress in Healthy Male Subjects, Rat Hepatocytes, and Recombinant NOS. Oxidative Medicine and Cellular Longevity Volume 2014, Article ID 212576, 12 pages; http://dx.doi.org/10.1155/2014/212576). However no patients needed to take this acute medication probably in relation to shortness of the study period, also confirming biofeeedback effectiveness in migraine treatment.
This informations were added in the Materials and Methods section and in the Results section.
Reviewer 2 Report
The aim of the present study is to verify the usage of an Artificial Neural Network (ANN) named ARIANNA, in the serum levels of the decomposition products of NO, nitrite and nitrate (NOx), the SOD serum levels, and the Migraine Disability Assessment Scores (MIDAS), could constitute prognostic variables predicting biofeedback efficacy in migraine treatment. Twenty women, affected by chronic migraine were enrolled and underwent an EMG-Biofeedback treatment. The results showed an accuracy of ANN of 75% in predicting the post treatment MIDAS score, highlighting a statistically significant correlation (R=-0.675, p=0.011) between NOx (nitrite and nitrate) and MIDAS only when peroxides were within a specific serum level range. In conclusion, ANN proves to be an innovative methodology to interpret the complex biological phenomena and biofeedback treatment in migraine.
However, the following minor points should be addressed to improve the quality of the work.
More recent literature on the topic should be included. Also, while discussing literature, present comparisons and contrasts of different studies.
Are there outlier in the data?
The authors should explain a bit how the data were collected?
The conclusion section must be enhanced by presenting comparison of the present study with existing studies.
Author Response
The aim of the present study is to verify the usage of an Artificial Neural Network (ANN) named ARIANNA, in the serum levels of the decomposition products of NO, nitrite and nitrate (NOx), the SOD serum levels, and the Migraine Disability Assessment Scores (MIDAS), could constitute prognostic variables predicting biofeedback efficacy in migraine treatment. Twenty women, affected by chronic migraine were enrolled and underwent an EMG-Biofeedback treatment. The results showed an accuracy of ANN of 75% in predicting the post treatment MIDAS score, highlighting a statistically significant correlation (R=-0.675, p=0.011) between NOx (nitrite and nitrate) and MIDAS only when peroxides were within a specific serum level range. In conclusion, ANN proves to be an innovative methodology to interpret the complex biological phenomena and biofeedback treatment in migraine.
However, the following minor points should be addressed to improve the quality of the work.
- More recent literature on the topic should be included. Also, while discussing literature, present comparisons and contrasts of different studies.
According to the Reviewer suggestion, we present and discuss additional recent literature about non-pharmacological approaches such as biofeedback.
Behavioural treatments guarantee significant results on classical primary endpoint as headache frequency and also in secondary endpoints as mood disorders, disability and quality of life acting on muscle contracture modulation, relaxation, biomarkers changes and on cortical excitability shutdown disrupting cortical spreading depression (Andrasik F, Grazzi L, Sansone E, D’Amico D, Raggi A, Grignani E. Non–pharmacological approaches for headache in young age: an update review. Frontiers in Neurology 2018;9;doi: 10.3389fneur.2018.01009). However, even if the use of biofeedback is supported by a conspicuous scientific evidences, the National Institute for Health and Care Excellence (NICE) of United Kingdom, does not recommended non-pharmacological treatments because most studies lack control groups and large sample sizes (National Clinical Guideline C National institute of Health and Clinical Excellence: Guidance. Headaches: Diagnosis and Management of Headaches in Young People and Adults. London: Royal College of Physicians (UK) National Guideline Centre; 2012 ). Anyway the majority of clinical studies show the efficacy of behavioural approaches on the headache treatment: in migraine prophylaxis, biofeedback shows the same effectiveness of pharmacological treatments, with additional effects when this approach is used in combination with pharmacological treatments; therefore behavioural approaches are useful in patients who cannot tolerate or are non-responsive to preventive or acute drugs or in childhood (Kroop P, Meyer B, Meyer W, Dresler T. an update on behavioural treatments in migraine – current knowledge and future options. Expert Review of Neurotherapeutics 2017, 17:1059-1068; Ailani J, Burch RC, Robbins MS, on behalf of the Board of Directors of the American Headache Society. The American Headache Society Consensus statement: update on integrating new migraine treatments into clinical practice. Headache 2021,61;1021-1039).
This additional text was added in the Introduction section
- Are there outlier in the data?
Authors confirm that no outliers are present in the data, as could be seen by the standard deviations reported in Table 1, all of them lower than 40% of mean values.
We modified sentence in the Results section: “Neither dropouts occurred during the study period, nor outliers were present in the data”.
- The authors should explain a bit how the data were collected?
As requested by the Reviewer we added other information about data collection, confirming that all elements of the study protocol was described in the mentioned previous manuscript of Ciancarelli et al. (Ciancarelli I, Tozzi-Ciancarelli MG, Spacca G, Di Massimo C, Carolei A. Relationship between biofeedback and oxidative stress in patients with chronic migraine. Cephalalgia. 2007;27:1136-41). Anyway we added in the text this sentence:” Collection of blood samples was performed to analyze nitrite and nitrate (NOx), SOD and peroxides; the first and the second blood sample were performed on two different days during a headache-free period, respectively, corresponding to the day of the first biofeedback session and of the last one”.
This sentence was added in the Material and Methods section.
- The conclusion section must be enhanced by presenting comparison of the present study with existing studies.
As suggested by the Reviewer we added comments describing a comparison of the present study with existing studies: “The results of our study are in line with the conclusions of the most recent manuscripts, confirming the efficacy of behavioural approaches on the headache treatment. Particularly, also our results showed that biofeedback, determining muscular relaxation and biomarkers modulation, represents an efficacy non-pharmacological approach for migraine prophylaxis, as described also in other manuscripts”.
This text was added in the Discussion section.
Reviewer 3 Report
The manuscript ID entitled "Identification of determinants of a biofeedback treatment efficacy in migraine by ARIANNA, an ARtificial Intelligent Assistant for Neural Network Analysis" is a good study. Migraine is a public health problem with severe socio-economic burden and related disability. Between non-pharmacological therapeutic approaches, behavioral treatments such as biofeedback have been proven effective for both adults and children. Oxidative stress is undoubtedly involved in migraine pathophysiology. Evidence showed a complex relationship between nitric oxide (NO) and superoxide anions (SOD), and their modifications after effective treatment. Conventional analyses may fail in highlighting complex, non-linear relationships among factors and outcomes. The aim of the present study was to verify if an Artificial Neural Network (ANN) named ARIANNA, might verify if the serum levels of the decomposition products of NO, nitrite, and nitrate (NOx), the SOD serum levels, and the Migraine Disability Assessment Scores (MIDAS), could constitute prognostic variables predicting biofeedback efficacy in migraine treatment. Twenty women, affected by chronic migraine were enrolled and underwent an EMG-Biofeedback treatment. The results showed an accuracy of ANN of 75% in predicting the post-treatment MIDAS score, highlighting a statistically significant correlation (R=-0.675, p=0.011) between NOx (nitrite and nitrate) and MIDAS only when peroxides were within a specific serum level range. In conclusion, ANN proves to be an innovative methodology to interpret the complex biological phenomena and biofeedback treatment in migraine. The manuscript is written in a good manner. I appreciate the authors for their great contribution. However, the following points are to be addressed,
- The chosen computational procedure was based on online training.----Not clear? how? any reference?
- Data from only 20 women and 5 variables------is it comfortable to access model performance?
- It is unclear "in which the above listed 30 variables are entered" how? the authors may include variables details
- How did the authors perform/ predicted Artificial Neural Network models? which platform? or web interface?
- "p-value" p must be italics
- In Table 1--- how does the author includes SEM values? is there any replication of the experiment?
- Discussion and conclusion may be improved.
- The conclusion may highlight the results and indicates the perspective of the study.
- Many typos and grammatical errors throughout the manuscript. The authors need to consider and improve the quality of the language as well.
Author Response
The manuscript ID entitled "Identification of determinants of a biofeedback treatment efficacy in migraine by ARIANNA, an ARtificial Intelligent Assistant for Neural Network Analysis" is a good study. Migraine is a The The manuscript ID entitled "Identification of determinants of a biofeedback treatment efficacy in migraine by ARIANNA, an ARtificial Intelligent Assistant for Neural Network Analysis" is a good study. Migraine is a public health problem with severe socio-economic burden and related disability. public health problem with severe socio-economic burden and related disability. Between non-pharmacological therapeutic approaches, behavioral treatments such as biofeedback have been proven effective for both adults and children. Oxidative stress is undoubtedly involved in migraine pathophysiology. Evidence showed a complex relationship between nitric oxide (NO) and superoxide anions (SOD), and their modifications after effective treatment. Conventional analyses may fail in highlighting complex, non-linear relationships among factors and outcomes. The aim of the present study was to verify if an Artificial Neural Network (ANN) named ARIANNA, might verify if the serum levels of the decomposition products of NO, nitrite, and nitrate (NOx), the SOD serum levels, and the Migraine Disability Assessment Scores (MIDAS), could constitute prognostic variables predicting biofeedback efficacy in migraine treatment. Twenty women, affected by chronic migraine were enrolled and underwent an EMG-Biofeedback treatment. The results showed an accuracy of ANN of 75% in predicting the post-treatment MIDAS score, highlighting a statistically significant correlation (R=-0.675, p=0.011) between NOx (nitrite and nitrate) and MIDAS only when peroxides were within a specific serum level range. In conclusion, ANN proves to be an innovative methodology to interpret the complex biological phenomena and biofeedback treatment in migraine. The manuscript is written in a good manner. I appreciate the authors for their great contribution. However, the following points are to be addressed,
- The chosen computational procedure was based on online training.----Not clear? how? any reference?
Authors now specified how online training works (also reporting references as required). We added a sentence in the Artificial Neural Network section: “online training uses information from one record at a time, updating the weights until one of the stopping rules is met [17-19]”
- Data from only 20 women and 5 variables------is it comfortable to access model performance?
The reviewer is right. Authors highlighted (and commented) it as the main limit of our study writing in the Discussion section:
“The main limit of this study was the small sample size of 20 participants, for each one 5 variables have been assessed at baseline. Further studies should investigate wider samples. However, despite this small dataset, the ANN achieved a good accuracy in predicting the outcome.”
- It is unclear "in which the above listed 30 variables are entered" how? the authors may include variables details
Authors are sorry because there was a misunderstanding about this sentence, that was simplified in the Artificial Neural Network section as follows: “ARIANNA is a multilayer perceptron, formed by the input layer, two hidden layers, and a final output layer (the output of which was the predicted outcome).”
All the information about variables are now reported in the following sentence in the Artificial Neural Network section:
“Differently from previous studies [17-19], the number of hidden units were not fixed, but auto-determined by the artificial neural network. The input layers referred to the following variables assessed pre-treatment: age, SOD, NOx, peroxides, MIDAS. The output layer provided the estimation of the outcome measure that was the MIDAS assessed post-treatment.”
- How did the authors perform/ predicted Artificial Neural Network models? which platform? or web interface?
As requested by the Reviewer we added this information in the Artificial Neural Network section:
“The ANN was implemented using the specific toolbox Neural Networks of IBM SPSS Statistics version 23 (Armonk, NY: IBM Corp).”
- "p-value" p must be italics
According to the Reviewer suggestion we made these changes in the Results section
- In Table 1--- how does the author includes SEM values? is there any replication of the experiment?
As requested by the Reviewer, Authors specified that in Table 1 we reported mean ± standard deviation and not the standard errors of the mean (we referred to the sample and not to the population). To clarify better the table we have also clarified that there is not any replication of the experiment, but the measures referred at pre- and post-treatment. The new legend of the Table 1 is the following: “Table 1. Mean ± standard deviation of the parameters assessed pre- and post-treatment for the group of participants, with the p-values of the paired comparison and normality check.”
- Discussion and conclusion may be improved.
Discussion was implemented commenting the most recent literature and comparing our results with different studies. This sentence was added in the Discussion section, also implementing bibliography: “The results of our study are in line with the conclusions of the most recent manuscripts, confirming the efficacy of behavioural approaches on the headache treatment. [2,4,6,14] Particularly, also our results showed that biofeedback, determining muscular relaxation and biomarkers modulation, represents an efficacy non-pharmacological approach for migraine prophylaxis, as described also in other manuscripts. [9,14, 15]”. Furthermore, we reported the major limit of our study, adding this sentence in the Discussin section: “The main limit of this study was the small sample size of 20 participants, for each one 5 variables have been assessed at baseline. Further studies should investigate wider samples. However, despite this small dataset, the ANN achieved a good accuracy in predicting the outcome”.
The conclusion section was improved with a sentence that described the perspective of our study; the sentence is reported as reply to point 8 .
- The conclusion may highlight the results and indicates the perspective of the study.
As suggested by the Reviewer, Authors added the following sentence in the Conclusion section: “In conclusion the perspective of this study is to reiterate the efficacy of biofeedback in the prophylactic treatment of migraine, and above all is to underline that the analyses of biological data with ANN may represent the appropriate methodology to identify the predicting factors of therapeutic effectiveness.”
- Many typos and grammatical errors throughout the manuscript. The authors need to consider and improve the quality of the language as well.
According to the reviewer suggestion, in this version we have carried out a careful revision of the English language.
Finally, the Authors jointly decided to change the title of the manuscript to better specify the contents of paper. Authors added the words “and oxidative stress”; therefore the modified title is: Identification of determinants of a biofeedback treatment efficacy in migraine and oxidative stress by ARIANNA, an ARtificial Intelligent Assistant for Neural Network Analysis
Authors kindly ask to Reviewers to evaluate and hopefully approve the new title.
Round 2
Reviewer 3 Report
Accept after minor revision (English language and style are fine/minor spell check required)
Author Response
Dear reviewer,
thank you. Authors agree with you. A professional English editing was performed. Attached the certification.
Best
the Authors
